# Socioeconomic Health Inequalities in Adolescent Metabolic Syndrome and Depression: No Mediation by Parental Depression and Parenting Style

**DOI:** 10.3390/ijerph18147716

**Published:** 2021-07-20

**Authors:** Alexander Lepe, Sijmen A. Reijneveld, Josué Almansa, Andrea F. de Winter, Marlou L. A. de Kroon

**Affiliations:** Department of Health Sciences, University Medical Center Groningen, University of Groningen, 9713 GZ Groningen, The Netherlands; s.a.reijneveld@umcg.nl (S.A.R.); j.almansa.ortiz@umcg.nl (J.A.); a.f.de.winter@umcg.nl (A.F.d.W.); m.l.a.de.kroon@umcg.nl (M.L.A.d.K.)

**Keywords:** health inequalities, adolescents, public health, epidemiology, cohort studies, socioeconomic status, metabolic syndrome, depression

## Abstract

We assessed to what extent parental depression and parenting style mediate the relationships between different measures of parental socioeconomic status (SES) and both depression and metabolic syndrome (MetS) in adolescents, and whether sex moderates these mechanisms. Data were from the prospective multigenerational Dutch Lifelines Cohort Study. Our sample consisted of 1217 adolescents with an average follow-up of 33.3 (SD = 7.33) months and a median baseline age of 13 (IQR:13–14) years. We used structural equation models to assess the direct and indirect effects of SES on baseline and changes at follow-up in both depression and MetS, and to assess moderation by sex. For each additional year of education, continuous MetS scores were 0.098 (95%CI: 0.020; 0.184) units lower at baseline and decreased 0.079 (95%CI: 0.004; 0.158) units at follow-up. No other direct or indirect effects of SES were found, and there was no moderation by sex. Additionally, warmer parenting style was generally associated with more favorable outcome scores. Therefore, improving parenting style may improve health for all adolescents. However, in this study parental depression and parenting style did not account for adolescent socioeconomic health inequalities. This may be partly due to good access to social services within the Netherlands.

## 1. Introduction

Socioeconomic inequalities exist across adolescents’ mental and physical health outcomes [1]. Reducing these inequalities would benefit the well-being and growth of both individuals and society [2]. To reduce these inequalities we need more evidence about the pathways that contribute to socioeconomic health inequalities, particularly for depression and metabolic syndrome (MetS), which differentially impact socioeconomically disadvantaged individuals [3,4] and also have long-term consequences into adulthood [5,6].

Depression and MetS are both important risk factors for cardiometabolic diseases [7,8], which, along with depression itself, are amongst the leading causes of disease burden [9]. Prevention of these conditions early in life would likely provide favourable consequences over the life course because depression is highly recurrent [5], and many features of MetS track from childhood into adulthood [6]. Moreover, MetS and depression may have a bidirectional relationship [10], so efforts to prevent one condition may benefit the other. In order to prevent these conditions, we need a better understanding of the mechanisms linking adolescent MetS and depression to socioeconomic status (SES).

Ample evidence links income to depression in adolescents via the Family Stress Model (FSM) or its components [3], but such evidence is lacking for MetS and the role of FSM remains understudied for other indicators of SES, e.g., education and occupation. The FSM links economic hardship to poor mental health outcomes in children through increased parental depression, marital problems, and disrupted parenting [3,11]. Studies which used components of the FSM have shown that parental depression and parenting style mediate the relationship between economic hardship and mental health [3,11]. Parental depression has been shown to be associated with childhood obesity [12], and similarly, parenting style mediated the relationship between SES and childhood trajectories of body mass index (BMI), which is another cardiometabolic risk factor [13]. However, these components of the FSM have not been investigated for MetS, and in general most studies on the FSM have not focused on the potentially unique roles of education and occupation [14].

More evidence is needed regarding difference in these mechanisms by sex. One study that used data from the US, specifically from African American families, found that males were more likely to experience depressive and anxious symptoms due to negative parent-adolescent relations compared to females [15]. However, another study conducted in a German population did not find a mediating effect of maternal negative communication for either sex [16]. To the best of our knowledge, no study has investigated the extent to which parental depression and parenting style mediate the relationship between SES and adolescent MetS and whether or not this differs by sex.

Evidence on the above-mentioned mechanisms may help to identify modifiable targets for preventive interventions to reduce socioeconomic health inequalities in adolescent depression and MetS. Therefore, we aimed to assess the extent to which parental depression and parenting style mediate the relationships between different measures of SES and depression and MetS at baseline, and the extent to which changes in the mediators influence changes in the outcomes. Because of the need for more evidence on the role of sex, we also investigated moderation by sex.

## 2. Methods

### 2.1. Setting and Population

Data were from the Lifelines Cohort Study, a multi-disciplinary prospective population-based cohort study examining in a unique three-generation design the health and health-related behaviours of 167,729 persons living in the North of The Netherlands [17,18]. It employs a broad range of investigative procedures in assessing the biomedical, socio-demographic, behavioural, physical and psychological factors which contribute to the health and disease of the general population, with a special focus on multi-morbidity and complex genetics. A detailed description of the recruitment strategy and data collection can be found elsewhere [17]. Briefly, Dutch-speaking individuals aged 25–49 were asked to participate by their physicians. Those who accepted were subsequently asked to invite their family members. Individuals could also self-register through the Lifelines website. As parenting style was assessed in participants aged 13–17 years during both assessments, we included all adolescents aged 13–17 years at baseline whose follow-up visit also occurred before adulthood (<18 years) (*n* = 2771). Our sample consisted of the 1217 (43.9%) individuals who participated in the second assessment with a mean follow-up duration of 33.3 (SD 7.33) months. They were recruited to join the Lifelines Cohort study by their parents who also participated in this cohort. Parental data was collected from all parents registered in Lifelines. For both adolescents and parents, data were collected during the baseline (2007–2014 and 2010–2014 in adults and children, respectively) and second assessment (2014–2018). Written informed consent was obtained for each participant prior to participating in the cohort. The Lifelines Cohort study is conducted according to the conventions set forth in the Declaration of Helsinki, and it has received approval from the Medical Ethics Committee of the University Medical Center Groningen (METc approval number: 2007/152).

### 2.2. Procedures

Participants completed questionnaires, physical exams, and venous blood draws during both the baseline and second assessment. Questionnaire data was self-reported and covered various topics including demographics. Physical exams and venous blood draws were conducted by trained research nurses using a standardised protocol [17].

### 2.3. Measures

Depressive symptoms were assessed using the anxious/depressed subscale of the Youth Self-Report (YSR). The YSR is a reliable and validated questionnaire used to measure emotional and behavioural problems [19]. The anxious/depressed subscale consists of 13 items, and adolescents indicate how often or to what extent each behaviour described in the item has occurred within the past six months on a scale from “not true” (0) to “very true, or often true” (2) [19]. These items were summed to create a score with possible values ranging from 0 to 26, which had a Poisson distribution. Higher scores indicate the presence of more depressive symptoms. A depression change score was created by subtracting baseline values from second assessment values.

MetS is a cluster of cardiometabolic risk factors (i.e., waist circumference, mean arterial pressure, insulin resistance, triglycerides, and high-density lipoproteins) [20]. Following an approach similar to Eisenmann et al. [20], each component was regressed on age and sex, and their residuals were standardised (see R code in Appendix A). Then, a continuous MetS (cMetS) score was built by summing the standardised residuals of all components. The standardised high-density lipoprotein residuals were reversed due to their inverse relationship with health. Eisenmann et al. [20] measured insulin resistance using the Homeostasis model assessment, but in our procedure we replaced this with fasting glucose, which is the measure for insulin resistance used in the Lifelines cohort. The cMetS score indicates how an individual’s cardiometabolic health compares to the rest of our sample [20]. Component scores at the second assessment were standardised with regression coefficients and residual standard deviations from the baseline assessment. Additionally, we calculated BMI standard deviation scores (SDS) as another proxy of cardiometabolic risk by first calculating BMI (kg/m^2^). Subsequently, we estimated SDS using Growth Analyser RCT version 4.1.5 (Rotterdam, The Netherlands) with an age- and sex-specific Dutch reference standard [21,22]. Change scores were created for both cMetS and BMI SDS by subtracting baseline values from second assessment values.

Parental depression was assessed using the Mini-International Neuropsychiatric Interview (MINI), a validated and reliable short structured diagnostic interview which evaluates the presence of either major depressive disorder (MDD) or dysthymia according to DSM-IV criteria [23]. Parental depression was defined as the presence of either MDD or dysthymia in at least one parent. Changes in parental depression were operationalized as ever having parental depression during either measurement wave.

Parenting style was measured using eight items from the Egna Minnen Betraffande Uppfostran [24]. Specifically, four items were taken from both the emotional warmth and rejection scales, assessing adolescents’ perceptions of their parents’ rearing practice. Responses to each item ranged from “never” (0) to “yes, almost always” (3). Sum scores were created for the warmth and rejection scales, which were combined by subtracting the warmth score from the rejection score. This resulted in a positively skewed scale with a possible range from −12 to 12. To fit a Poisson distribution, we added 12 to all scores, resulting in a possible range from 0 (complete emotional warmth) to 24 (complete rejection). The Cronbach’s alpha of this scale during the baseline and second assessment was 0.82 and 0.84 for mothers and 0.80 and 0.79 for fathers, respectively. Because mothers usually spend more time parenting [25], we constructed the parenting style variable for both assessments using the mother’s score. If this was missing, the father’s score was used. A parenting style change score was created by subtracting baseline values from second assessment values.

SES was measured using three separate indicators: parental education, occupation, and income. This data was obtained from both parents during the baseline assessment. Education was assessed by asking parents about the highest educational level they attained, with eight potential responses ranging from “no education” to “university”. In an approach similar to De Graaf et al. [26], these categories were recoded into years of education using the number of years it would take to complete each category by the fastest route possible; for example, no education, primary school, secondary school, and university were coded as 5, 6, 12, and 16 years, respectively. Occupation was coded using the International Standard Classification of Occupations 2008 [27]. This was recoded into Treiman’s Standard International Occupational Prestige Scale, which is a continuous measure of occupational prestige [28,29]. It focuses on the prestige an occupation gives its holder, not on the incomes associated with occupations. For both education and occupation, the highest level from either parent was used. If only one parent was registered in Lifelines, then data from that parent was used. To construct the measure of equivalised household income (income), net household income in Euros was divided by the square root of the number of people living on this income [30]. Similarly to how parenting style was constructed, the mother’s response was used, but the father’s response was used if this was missing.

### 2.4. Statistical Analysis

First, we described the characteristics of the sample during both assessments. We compared the baseline characteristics of those who participated in the second assessment and those lost to follow-up. For this comparison, we excluded adolescents aged 16 years or older at baseline from the lost to follow-up group, as they likely would have been excluded from our analysis due to being an adult at the second assessment.

To assess the direct and indirect, i.e., mediated, effects of SES on both baseline values and changes (Figure 1a,b) in depression and MetS, we used structural equation modelling. The baseline model assumed a probit distribution for parental depression and a Poisson distribution for parenting style and adolescent depression. The changes model assumed a probit distribution for parental depression, and all other variables were assumed to be normally distributed. The two outcome variables were allowed to correlate. For this, we added a random intercept to the Poisson regression for adolescent depression at baseline, which correlated with the residuals of cMetS. Confidence intervals for the path coefficients were estimated non-parametrically, with 1000 bootstrap replications.

To assess moderation by sex, we ran multigroup analyses and tested if the path coefficients differed by sex. Missing data was assumed to be missing at random, and was accounted for by estimating all models using full information maximum likelihood. We also conducted sensitivity analyses using BMI SDS instead of cMetS for all models. In total, including the sensitivity analyses, 12 models were estimated. The structural equation modelling was performed using Mplus v8.4 (Muthen & Muthen, Los Angeles, LA, USA) [31]. All other data preparation and analyses were conducted using R version 3.5.2 (R Foundation for Statistical Computing, Vienna, Austria) [32].

## 3. Results

### 3.1. Sample Characteristics

Table 1 provides a summary of the characteristics of the adolescents included in the analyses. There were only two significant differences between females and males. On average, females’ cMetS scores decreased more over time than males’ cMetS score (−0.76 vs. 0.10, respectively; *p* < 0.001). Furthermore, the median number of symptoms of depression at baseline was higher in females than in males (3 vs. 1, respectively; *p* < 0.001). There were no other significant differences between females and males. They both generally came from households with emotionally warm parenting, and the baseline prevalence of parental depression was relatively low (3.2%) for both sexes. Adolescents lost to follow-up were older, had slightly worse biomarkers, and came from slightly less emotionally warm households (Table 2).

### 3.2. SES and Baseline Depression and cMetS: Mediation by Baseline Parental Depression and Parenting Style

Only parental education had an effect on the outcomes, and there were no indirect effects (Table 3). Parental education only had a total and direct effect on cMetS; the direct effect resulted in a 0.098 units (95% confidence interval (CI): 0.020; 0.184) decrease in cMetS per additional year of parental education. Adolescents whose parents completed university education on average had cMetS scores that were lower than children whose parents only completed secondary school (0.39 units) or primary school (0.98 units). Additionally, less emotionally warm parenting was consistently associated with higher adolescent depression and cMetS scores. No sex differences were found for either education (*p* = 0.084), income (*p* = 0.358), or occupation (*p* = 0.634). The sensitivity analyses which included BMI SDS (Appendix A) yielded similar results to the models using MetS (Appendix A).

### 3.3. SES and Changes in Depression and cMetS: Mediation by Changes in Both Parental Depression and Parenting Style

Again, parental education only had a direct effect on changes in cMetS, and there were no indirect effects (Table 4). The direct effect of education on changes in cMetS resulted in a 0.079 (95%CI: 0.004; 0.158) units decrease at the second assessment, per additional year of parental education. Adolescents whose parents completed university education on average had cMetS change scores that were lower than adolescents whose parents only completed secondary school (0.316 units) or primary school (0.790 units). Additionally, parenting which became less emotionally warm was consistently related to increases in symptoms of depression during follow-up. No sex differences were found in the models including either education (*p* = 0.293), income (*p* = 0.624), or occupation (*p* = 0.649). The sensitivity analyses which included BMI SDS (Appendix A) yielded similar results to the models using MetS (Appendix A).

## 4. Discussion

We assessed to what extent parental depression and parenting style mediate the relationships between SES and both depression and MetS, and if these mechanisms differed by sex. We found that parental depression and parenting style did not mediate these relationships, and they also did not differ by sex. While parenting style did not account for the relationship between SES and our outcomes, it was generally related to our outcomes.

Each indicator of SES had a unique relationship with the mediators and the outcomes, which was in line with our expectations [14]. Higher parental SES was generally associated with less parental depression, but only parental education was associated with adolescent cMetS. The latter contrasts with our expectations based on the FSM studies demonstrating a relationship between parental income and adolescent health [3]. An explanation for our contradictory findings may be that the FSM typically focuses on economic hardship, whereas we used income as a proxy of SES [14]. In the Netherlands, social assistance is considered a right meant to provide a minimum subsistence level [33]; this includes equal access to education and care. This may mitigate the negative effects of low income on adolescent health.

Parental depression was predicted by income and ever parental depression was predicted by all indicators of SES, but ever parental depression was only associated with changes in BMI; none of the resulting mediating pathways were statistically significant. A recent review [3] identified two studies which reported that parental depression mediated the relationship between SES and adolescent depression [15,34]. These studies used samples from the US, of which one consisted of African American families with 40% living below the poverty line [15], and the other consisted of both Caucasian and African American families with an oversampling of African American and low income families. Comparatively, our sample was predominately Caucasian and had an overrepresentation of highly educated individuals, which implies that our sample had less particularly disadvantaged individuals [35]. Moreover, the US and the Netherlands also have very dissimilar social safety nets, which may partially account for differences in the results. Lastly, the other studies used measures which included the number of parental depression symptoms. This may have allowed them to capture important effects of increases in subclinical depressive symptoms, and may have increased their statistical power.

Parenting style was predicted by education and occupation, and was associated with lower depression and MetS scores. However, it did not significantly mediate any of the socioeconomic gradients, and we did not find any differences by sex. This contrasts with a previous study which found that relationships with parents mediated the association between economic hardship and adolescent depression, and this relationship differed by sex [15]. Possible reasons for these differences are that we defined parenting style using a scale based on warmth and rejection, while the aforementioned study used a measure based on aggression and conflict, and differences in the social circumstances between samples. This is supported by a German study, which is more culturally and socially similar than the US sample. In the German study, parenting did not mediate the relationship between economic deprivation and adolescent depression, and the role of parenting was similar in both males and females [16].

### 4.1. Strengths and Limitations

A major strength of this study is its community-based and longitudinal design which allowed us to assess changes in our mediators and outcomes in a sample generally representative of this region of the Netherlands [35]. We also used multiple indicators of SES to account for their unique relationships with health [14]. Lastly, our results accounted for the bi-directional relationship between cMetS and depression.

This study also has limitations. First, the adolescents lost to follow-up come from households with slightly worse parenting styles (Table 2). By not accounting for these individuals, we are effectively conditioning our analyses on having slightly better parenting style, and this would prevent a proper estimation of the indirect effect of SES on our outcomes. Additionally, we could not account for pubertal status, which is known to influence cardiometabolic risk factors and depression [36,37]. Lastly, our measure of depression also captures anxiety symptoms, which may have introduced some measurement error.

### 4.2. Implications

Our findings suggest that neither parental depression nor parenting style are good intervention targets to reduce socioeconomic health inequalities amongst Dutch adolescents. However, parenting style was consistently associated with our outcomes, making this an interesting cue for interventions. The effects of parenting style may be considerable, e.g., for every unit decrease in warm parenting at baseline, cMetS scores increased by 0.58 units. Moreover, for each unit decrease in baseline parenting style, depression scores increased by 2.8%, and for each unit decrease in the parenting style change score, depression scores increased by 12.8% during follow-up. Interventions aimed at improving parenting style would likely result in a decrease of both depression and MetS in adolescents.

Our study also demonstrates the need for further studies to investigate which factors may account for socioeconomic health inequalities in Dutch adolescents. For instance, parental health literacy may be an important modifiable factor which has been understudied in this age group. It is well-known that SES is associated with health literacy in adults, and that parental health literacy is associated with children’s health [38]. However, there is a lack of research assessing the extent to which parental health literacy mediates this pathway [39].

## 5. Conclusions

We found that neither parental depression nor parenting style accounted for socioeconomic health inequalities amongst adolescents, and these pathways did not differ according to sex. However, parenting style did play an important role in the health of these adolescents. Therefore, improving parenting style may help reduce the overall burden of depression and MetS. Lastly, additional studies are needed to identify which mechanisms account for socioeconomic differences in MetS in Dutch adolescents, so that these factors can be targeted in preventive interventions at a national level.

## Figures and Tables

**Figure 1 ijerph-18-07716-f001:**
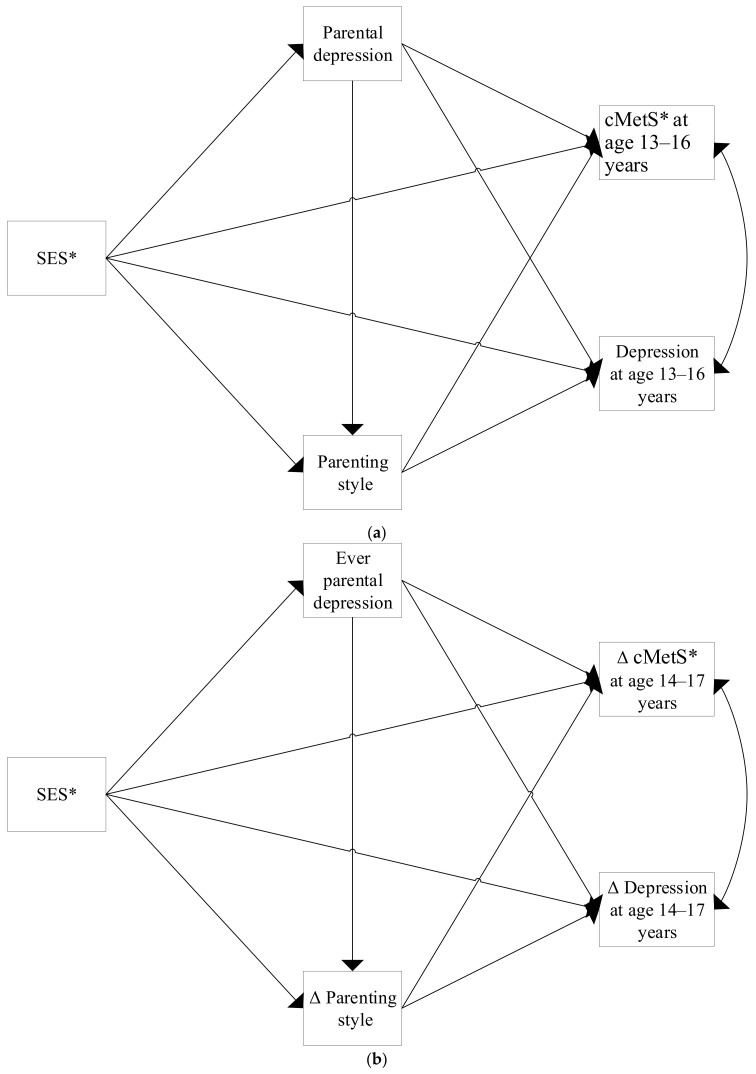
Theoretical model demonstrating the direct and indirect pathways connecting SES to both (**a**) baseline values and (**b**) changes in depression and MetS. * Separate models were estimated for each proxy of SES (education, income, and occupation). Additionally, sensitivity analyses were conducted using BMI and ΔBMI instead of cMetS and ΔcMetS, respectively.

**Table 1 ijerph-18-07716-t001:** Description of the characteristics of the study population used in the analyses (*n* = 1217), stratified by sex. The number of participants and percentages are shown for categorical variables, means and standard deviations are given for normally distributed continuous variables, and the median and interquartile range are provided for non-normally distributed continuous variables.

	Males (*n* = 589)		Females (*n* = 628)		
	Mean (sd), Median [IQR], or *n* (n%)	Missing n (n%)	Mean (sd), Median [IQR], or *n* (n%)	Missing *n* (n%)	*p*-Value ^e^
Baseline Age (years)	13 [13–14]		13 [13–14]		0.192
Health Outcomes					
Continuous MetS score	0.03 (2.89)	64 (10.9%)	0.02 (2.79)	75 (11.9%)	0.956
Δ ^a^ Continuous MetS score	0.10 (2.49)	117 (19.9%)	−0.76 (2.40)	135 (21.5%)	<0.001
Symptoms of depression	1 [0–3]	19 (3.2%)	3 [1–5]	24 (3.8%)	<0.001
Δ ^a^ Symptoms of depression	0.09 (2.76)	157 (26.7%)	0.00 (3.44)	132 (21.0%)	0.812
Baseline Mediators					
Parental depression	19 (3.2%)	14 (2.4%)	20 (3.2%)	19 (3.0%)	1.000
Parenting style	3 [1–5]	21 (3.6%)	3 [1–5]	25 (4.0%)	0.980
Changes Mediators					
Ever parental depression ^b^	36 (6.1%)	87 (14.8%)	45 (7.2%)	89 (14.2%)	0.490
Δ ^a^ Parenting style	−0.12 (3.66)	147 (25.0%)	−0.39 (3.31)	111 (17.7%)	0.289
SES					
Years of education	12 [12–15]	6 (1.0%)	12 [12–15]	0 (0%)	0.367
Equivalized household income ^c^	1375 [1125.0–1677.1]	57 (9.7%)	1375 [1010.4–1625.0]	82 (13.1%)	0.056
Occupation ^d^	49.13 (12.11)	1 (0.2%)	48.16 (12.70)	6 (1.0%)	0.162

^a^ Δ: Change measured by subtracting the baseline measurement from the second assessment measurement ^b^ Ever Parental Depression: Indicates the presence of depression at either the baseline or second assessment ^c^ Equivalized household income: Calculated as the net household income in Euros divided by the square root of the number of individuals who live off of the income. ^d^ Occupation: Measured using the standard international occupational prestige scale, which is a continuous measure of occupation. It focuses on the prestige an occupation gives its holder, not on the incomes associated with occupations. ^e^ *p*-values from testing the difference between the two groups using Fisher’s exact test for categorical variables or Kruskal-Wallis test for numerical variables.

**Table 2 ijerph-18-07716-t002:** Description of the study population at baseline, stratified by those who participated in the second assessment and those who were lost to follow-up and were less than 16 years old.

		Participated during 2A (*n* = 1217)		Lost to Follow-Up (*n* = 945)		
		Mean (sd), Median [IQR], or *n* (n%)	Missing *n* (n%)	Mean (sd), Median [IQR], or *n* (n%)	Missing *n* (n%)	*p*-Value ^c^
Sex	Male	589 (48.4%)		461 (48.8%)		0.862
	Female	628 (51.6%)		484 (51.2%)		
Age (years)		13 [13–14]		14 [13–15]		<0.001
MetS Components:						
Fasting Glucose (mmol/L)		4.73 (0.46)	135 (11.1%)	4.72 (0.48)	162 (17.1%)	0.793
HDL (mmol/L)		1.52 (0.32)	106 (8.7%)	1.48 (0.30)	121 (12.8%)	0.001
Triglycerides (mmol/L)		0.66 [0.51–0.88]	106 (8.7%)	0.68 [0.52–0.90]	121 (12.8%)	0.145
Mean Arterial Pressure (mmHg)		80.02 (6.29)	5 (0.4%)	80.58 (6.34)	12 (1.3%)	0.050
Waist Circumference (cm)		70.54 (8.07)	3 (0.2%)	72.28 (8.11)	11 (1.2%)	<0.001
Health Outcomes						
BMI SDS		0.38 (1.07)	3 (0.2%)	0.50 (1.05)	11 (1.2%)	0.023
Symptoms of depression		2 [1–4]	43 (3.5%)	2 [0–4]	54 (5.7%)	0.882
Baseline Mediators						
Parental Depression		39 (3.2%)	33 (2.7%)	33 (3.5%)	26 (2.8%)	0.718
Parenting Style		3 [1–5]	46 (3.8%)	3 [1–6]	57 (6.0%)	0.015
SES						
Years of education		12 [12–15]	6 (0.5%)	12 [12–15]	9 (1.4%)	0.146
Equivalized household income ^a^		1375 [1125–1677.1]	139 (11.4%)	1375 [1039.4–1638.0]	101 (10.7%)	0.170
Occupation ^b^		48.63 (12.42)	7 (0.6%)	47.86 (12.73)	14 (1.5%)	0.227

^a^ Equivalized household income: Calculated as the net household income in Euros divided by the square root of the number of individuals who live off of the income. ^b^ Occupation: Measured using the standard international occupational prestige scale, which is a continuous measure of occupation. It focuses on the prestige an occupation gives its holder, not on the incomes associated with occupations. ^c^ *p*-values from testing the difference between the two groups using Fisher’s exact test for categorical variables or Kruskal-Wallis test for numerical variables.

**Table 3 ijerph-18-07716-t003:** The estimated effects of SES, parental depression, and parenting style on depression and cMetS at baseline. Statistically significant effects (*p* < 0.05) are bolded. Estimates for all the paths shown in Figure 1a can be seen in Appendix A.

	Depression Coefficient [95% CI]	cMetS Coefficient [95% CI]
Education ^b^		
Direct effect of parental depression	0.185 [−0.284; 0.552]	−0.122 [−0.961; 0.755]
Direct effect of parenting style	**0.028 [0.008; 0.047]**	**0.056 [0.003; 0.107]**
Direct effect of education	0.022 [−0.009; 0.052]	**−0.098 [−0.184; −0.020]**
Education via parental depression	−0.033 [−0.156; 0.039]	0.021 [−0.123; 0.207]
Education via parenting style	−0.001 [−0.002; 0.000]	−0.002 [−0.004; 0.000]
Education via parental depression and parenting style	−0.001 [−0.004; 0.000]	−0.003 [−0.010; 0.001]
Total effect ^a^ of education	0.018 [−0.013; 0.049]	**−0.104 [−0.187; −0.026]**
Income ^c^		
Direct effect of parental depression	0.187 [−0.291; 0.541]	−0.045 [−0.924; 0.803]
Direct effect of parenting style	**0.028 [0.008; 0.048]**	**0.060 [0.007; 0.111]**
Direct effect of income	0.007 [−0.008; 0.023]	−0.004 [−0.044; 0.040]
Income via parental depression	−0.026 [−0.083; 0.038]	0.006 [−0.124; 0.127]
Income via parenting style	0.000 [−0.001; 0.000]	−0.001 [−0.003; 0.000]
Income via parental depression and parenting style	−0.001 [−0.003; 0.000]	−0.002 [−0.007; 0.001]
Total effect ^a^ of income	0.006 [−0.011; 0.020]	−0.008 [−0.051; 0.032]
Occupation ^d^		
Direct effect of parental depression	0.172 [−0.299; 0.523]	−0.049 [−0.931; 0.783]
Direct effect of parenting style	**0.028 [0.007; 0.047]**	**0.059 [0.007; 0.110]**
Direct effect of occupation	0.023 [−0.025; 0.071]	−0.057 [−0.188; 0.075]
Occupation via parental depression	−0.033 [−0.135; 0.069]	0.009 [−0.156; 0.222]
Occupation via parenting style	−0.002 [−0.004; 0.000]	−0.003 [−0.009; 0.000]
Occupation via parental depression and parenting style	−0.002 [−0.006; 0.000]	−0.003 [−0.012; 0.001]
Total effect ^a^ of occupation	0.018 [−0.031; 0.065]	−0.066 [−0.201; 0.067]

^a^ Total effects are estimated using models where all of the mediators are removed, which is why it is not simply the sum of the direct and indirect effects with non-normal outcomes. ^b^ Education: Measured using years of education. ^c^ Equivalized household income: Calculated as the net household income in Euros divided by the square root of the number of individuals who live off of the income; the results have been scaled to per 100 Euros. ^d^ Occupation: Measured using the standard international occupational prestige scale, which is a continuous measure of occupation. It focuses on the prestige an occupation gives its holder, not on the incomes associated with occupations. The results have been scaled to per 10 units of prestige.

**Table 4 ijerph-18-07716-t004:** The estimated effects of SES, parental depression, and parenting style on changes in depression and cMetS. Statistically significant effects (*p* < 0.05) are bolded. Results for all the paths shown in Figure 1b can be seen in Appendix A.

	Δ Depression Coefficient [95% CI]	Δ cMetS Coefficient [95% CI]
Education ^b^		
Direct effect of parental depression	−0.390 [−1.046; 0.293]	0.034 [−0.609; 0.738]
Direct effect of parenting style	**0.120 [0.058; 0.179]**	−0.002 [−0.046; 0.042]
Direct effect of education	0.072 [−0.013; 0.149]	**−0.079 [−0.158; −0.004]**
Education via ever parental depression	0.078 [−0.051; 0.261]	−0.007 [−0.156; 0.127]
Education via Δ parenting style	−0.002 [−0.014; 0.010]	0.000 [−0.002; 0.002]
Education via ever parental depression and Δ parenting style	0.016 [−0.002; 0.043]	0.000 [−0.009; 0.008]
Total effect ^a^ of education	0.079 [−0.008; 0.159]	**−0.080 [−0.161; −0.005]**
Income ^c^		
Direct effect of parental depression	−0.369 [−1.049; 0.345]	0.069 [−0.591; 0.755]
Direct effect of parenting style	**0.121 [0.059; 0.180]**	−0.003 [−0.047; 0.041]
Direct effect of income	0.016 [−0.030; 0.061]	−0.016 [−0.057; 0.022]
Income via ever parental depression	0.041 [−0.037; 0.139]	−0.008 [−0.091; 0.069]
Income via Δ parenting style	−0.003 [−0.010; 0.004]	0.000 [−0.002; 0.002]
Income via ever parental depression and Δ parenting style	0.009 [−0.001; 0.025]	0.000 [−0.005; 0.004]
Total effect ^a^ of income	0.018 [−0.029; 0.064]	−0.016 [−0.055; 0.022]
Occupation ^d^		
Direct effect of parental depression	−0.372 [−1.043; 0.348]	0.093 [−0.559; 0.779]
Direct effect of parenting style	**0.120 [0.058; 0.181]**	−0.002 [−0.046; 0.041]
Direct effect of occupation	0.028 [−0.114; 0.159]	−0.073 [−0.187; 0.039]
Occupation via ever parental depression	0.114 [−0.095; 0.362]	−0.028 [−0.248; 0.184]
Occupation via Δ parenting style	−0.002 [−0.022; 0.018]	0.000 [−0.004; 0.004]
Occupation via ever parental depression and Δ parenting style	0.025 [−0.002; 0.069]	0.000 [−0.013; 0.011]
Total effect ^a^ of occupation	0.048 [−0.095; 0.181]	−0.076 [−0.190; 0.040]

^a^ Total effects are estimated using models where all of the mediators are removed, which is why it is not simply the sum of the direct and indirect effects. ^b^ Education: Measured using years of education. ^c^ Equivalized household income: Calculated as the net household income in Euros divided by the square root of the number of individuals who live off of the income; the results have been scaled to per 100 Euros. ^d^ Occupation: Measured using the standard international occupational prestige scale, which is a continuous measure of occupation. It focuses on the prestige an occupation gives its holder, not on the incomes associated with occupations. The results have been scaled to per 10 units of prestige.

## Data Availability

Researchers can apply to use the Lifelines data used in this study. More information about how to request Lifelines data and the conditions of use can be found on their website (https://www.lifelines.nl/researcher/how-to-apply) (accessed on 18 June 2021).

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
