# Peer review of "Socioeconomic Health Inequalities in Adolescent Metabolic Syndrome and Depression: No Mediation by Parental Depression and Parenting Style"

_ijerph, 2021, doi:10.3390/ijerph18147716_

Round 1
Reviewer 1 Report
The manuscript "Socioeconomic Health Inequalities in Adolescent Metabolic Syndrome and Depression: no mediation by parental depression and parenting style" explores the possible mediating effect of parental depression and parenting style on the relationship between SES and both depression and MetS among Dutch adolescents. This is an interesting topic, but I have some reservations about the contribution of this article in general. More work needs to be done to balance the quality of all sections of the manuscript.
The sections that need the most modifications are mainly the Introduction and the description of the Results.
-Introduction. Since this is the first section the reader reads, it is here that the authors should clearly explain the object of study. The current form of this section does not clearly explain what the authors want to analyze and why. Here is an example of two confusing sentences due to repetition of concepts and misuse of the semicolon:
"Depression is a leading cause of disease burden, and both depression and MetS are important risk factors for cardiometabolic diseases [7,8], which are also amongst the leading causes of disease burden [9]. Prevention of these conditions early in life would likely provide favourable consequences over the life course; depression is highly recurrent [3], and many features of MetS track from childhood into adulthood [4]." (Second paragraph, first page)
In addition, the authors state in line 56 that there is a conflict in previous literature about the mechanisms by sex. However, the conflict between the two references provided by the authors could be originated by the distinct reference population in both studies (reference 15 focused on urban African American families).
Overall, this section requires a deep revision.
Results:
-The authors must provide more details about the characteristics of the working sample as well as of the models (Tables 3 and 4). The current version of this section is too succinct.
Minor issues:
-it must be included the link in reference 9
Author Response
Dear Reviewer,
We thank you very much for help and your clarifying comments. We have addressed the issues you have raised in the attached document. Our responses are in red, and any text that we added to the manuscript is shown in italics within our uploaded response.
With kind regards, also on behalf of the coauthors,
Alexander Lepe

Reviewer 2 Report
A unique study due to its embedding in a large cohort study and combining laboratory and questionnaire results. An important aspect of the inclusion of social factors as a root cause of cardiovascular disease was also addressed.
The methodology of the study and the presentation are very clear.
The main shortcoming is the imprecise description of the METs variable. The cited paper [20] talks about the number of risk factors and a dichotomous final outcome (for children 7-9 years). It is not clear whether this is the author's approach to obtain a standardised continuous METs. variable. The mean is about zero, however SD >2. A taxonomic method rather than a z-score?. Many centres would be interested in replicating this method of analysis. I suggest an additional appendix with a description of the construction of this indicator or please provide another reference.
I suggest also some minor clarifications.
1) It would be better to explain the timing of the initiation of the Lifelines Cohort Study, which seems to be a wonderful undertaking. On what principle were children included in the study. However the year of the study is given elsewhere in the description by the blood test.
2) The convention for describing SES could be standardised. At figure 1 it is worth clarifying in a footnote whether these are three models for: occupation, years of education, equalized household income. Actually in the appendix there are 12 models, also models for BMI and deltaBMI.
3) It could be more clearly stated that 12 models were estimated
4) Full descriptions of the 3 SES variables could be introduced into first column of Tables 3 and 4 to emphasise that the type of variables is according to the assumptions of the SEM models.
5) The description of SES in the methods (lines 142+) should be more consistent with the footnote to Tables 1 and 2.
Author Response

(The authors gave the same response as above.)

Reviewer 3 Report
Comments and Suggestions
The article is well structured in all its parts.
This article talk about Socioeconomic Health Inequalities in Adolescent Metabolic Syndrome and Depression . The goal of these article is to evaluate the extent to which parental depression and parenting style mediate the relationships between different measures of SES, Depression and Mets .
Comments:
- line 93, there is an error. delete the word "procedures"
- Page 5, I suggest to increase the font size in the text of the figure 1, and to extend the description of the caption.
- From line 184 to 188 the description of the Table 1 is very general. I suggest expanding the description and referring to numerical data.
- In line 236 I think there is an Error. Please delete the word "Author"
- From line 310 to 315, the Conclusion are very short. There is no exhaustive description of the result obtained, there are no numerical data. I suggest expanding this section with more detailed informations.
Author Response

(The authors gave the same response as above.)

Round 2
Reviewer 1 Report
The manuscript has improved significantly. I have no further comments. I just want to congratulate the authors.